# Solvent Free Three-Component Synthesis of 2,4,5-trisubstituted-1*H*-pyrrol-3-ol-type Compounds from *L*-tryptophan: DFT-B3LYP Calculations for the Reaction Mechanism and 3*H*-pyrrol-3-one↔1*H*-pyrrol-3-ol Tautomeric Equilibrium

**DOI:** 10.3390/molecules25194402

**Published:** 2020-09-25

**Authors:** Diego Quiroga, Lili Dahiana Becerra, Ericsson Coy-Barrera

**Affiliations:** Bioorganic Chemistry Laboratory, Facultad de Ciencias Básicas y Aplicadas, Universidad Militar Nueva Granada, Campus Nueva Granada, Cajicá 250247, Colombia; lilidahiana18@gmail.com (L.D.B.); ericsson.coy@unimilitar.edu.co (E.C.-B.)

**Keywords:** 1*H*-pyrrol-3-ol, enamines, *L*-tryptophan, indole phytoalexin, hybrid heterocycles

## Abstract

In this paper, we describe the solvent-free three-component synthesis of 2,4,5-trisubstituted-1*H*-pyrrol-3-ol-type compounds from *L*-tryptophan. The first step of the synthetic methodology involved the esterification of *L*-tryptophan in excellent yields (93–98%). Equimolar mixtures of alkyl 2-aminoesters, 1,3-dicarbonyl compounds, and potassium hydroxide (0.1 eq.) were heated under solvent-free conditions. The title compounds were obtained in moderate to good yields (45%–81%). Density functional theory using “Becke, 3-parameter, Lee–Yang–Parr” correlational functional (DFT-B3LYP) calculations were performed to understand the molecular stability of the synthesized compounds and the tautomeric equilibrium from 3*H*-pyrrol-3-one type intermediates to 1*H*-pyrrol-3-ol type aromatized rings.

## 1. Introduction

The five-membered heterocyclic systems occur in a large number of natural secondary (currently called specialized) metabolites. These compounds usually exhibit a wide variety of biological and biomedical properties [1]. Within this type of compounds, 1*H*-pyrrole is representative since it is well-known as a fundamental structural subunit in many of the naturally-occurring biological agents [2]. From this fact, many synthetic pyrrole derivatives have been obtained on the basis of biomimetic exercises, and have shown promissory behavior as antipsychotics, adrenergic antagonists, anxiolytics, anticancer, antibacterial and antifungal agents [3,4,5]. For example, synthetic 1,5-diphenylpyrrole-type compounds have shown high activity against a panel of *Enterococcus faecium, Staphylococcus aureus, Klebsiella pneumoniae, Acinetobacter baumannii, Pseudomonas aeruginosa y Enterobacter cloacae* (ESKAPE) bacteria, similar to or even better than levofloxacin, a widely known antibiotic [6]. In addition, the antimicrobial activity of 1*H*-pyrrole-2-carboxylic acid and derivatives have been reported. Such a compound was identified as a secondary metabolite of *Streptomyces griseus*, considered the main bioactive compound against the oomycete *Phytophthora capsici* [7]. On the other hand, a series of 2-alkyl-4-bromo-5-(trifluoromethyl)pyrrole-3-carbonitriles was obtained having excellent fungicidal activities against *Alternaria solani* and *Fusarium oxysporum* [8]. Substituted 3,4-dimethyl-1*H*-pyrrole-2-carboxamides and carbohydrazide analogues could be considered potential antifungal and antimicrobial agents, especially against the pathogen *Aspergillus niger* [9].

Several studies employing hybrid heterocyclic systems have led to a synergistic effect, expanding the scope on their biological activities [10,11,12,13]. Novel derivatives of pyrano[2,3-*b*]pyridine and pyrrolo[2,3-*b*]pyrano[2,3-*d*]pyridine proved to be active against *Candida sp.*, *Aspergillus multi*, *Aspergillus niger*, *Escherichia coli* and *Staphylococcus aureus*. Ethyl 4-methyl-1,7,8,9-tetrahydropyrano[2,3-*b*]pyrrolo[2,3-*d*]pyridine-3-carboxylate was found as the most active compound against all the mentioned microorganisms [14]. Other hybrid heterocyclic systems, such as pyrazole-furan carboxamide and pyrazole-pyrrole carboxamide-type compounds, have shown activities against three destructive fungi, including *Sclerotinia sclerotiorum*, *Rhizoctonia solani*, and *Pyricularia grisea* [15]. These studies suggested that this kind of heterocyclic systems can be considered phytoalexin-like analogues with promising bioactivity.

According to this background, the Bioorganic Laboratory at Universidad Militar Nueva Granada (UMNG) has deepened the synthesis and antifungal activity against the phytopathogenic fungus *Fusarium oxysporum* of novel 2,4,5-trisubstituted-1*H*-pyrrol-3-ol-type compounds **1**. To establish a high yielding and novel methodology, a comprehensive literature survey was then performed. Such a review ruled that the pyrrole nucleus can be obtained employing cycloaddition, cyclocondensation and carbon-carbon coupling reactions by metalorganic-catalyzed reactions [16]. All the described synthetic methods are diverse in terms of the employed reactants as well as the variation in the reaction conditions, comprising several representatives, such as Knorr, Hantzsch and Feist methodologies [16,17,18]. Some advantages have been achieved, primarily establishing more efficient and greener synthetic protocols [19,20,21]. Thus, the preparation of 3-substituted-4-(3-chloro-4-fluorophenyl)-1*H*-pyrrole derivatives based on the Van Leusen pyrrole synthesis was performed through a metal-free synthetic method, involving a C-N bond cleavage under mild reaction conditions [22]. The synthesis of 1,3,4-trisubstituted pyrroles via three-component domino reactions between (*E*)-3-(dimethylamino)-1-arylprop-2-en-1-ones, anilines and 2-nitrostyrenes was also reported. These reactions proceeded via addition–elimination/Michael addition/intramolecular annulation/elimination domino sequential reactions, with one C-C and two C-N bonds formation in a single synthetic operation [23]. Phenylpyrrole-substituted tetramic acid derivatives bearing carbonates were designed and synthesized from 4-(2,4-dioxopyrrolidin3-ylidene)-4-(phenylamino)butanoic acids [24]. An efficient, one-pot synthetic procedure for [4-(*tert*-butyl)-1*H*-pyrrol-3-yl](phenyl)methanone was proposed using acetophenone and trimethylacetaldehyde in the presence of toluenesulfonylmethyl isocyanide (TosMIC) and mild base [25]. On the other hand, an efficient, environmentally benign and molecular iodine-promoted protocol was recently described for the cascade synthesis of dihydro-1*H*-pyrrol derivatives. This methodology employed a four-component reaction between diethyl acetylenedicarboxylate (DEAD), aromatic amines and phenylglyoxal monohydrate catalyzed by 10 mol% iodine in ethanol at room temperature. The proposed reaction mechanism involved transformations through an imine intermediate followed by iodine-induced Mannich reaction and subsequent intramolecular cyclization sequences. Moreover, this protocol involved the formation of C-N, C-C and O-H bonds [26]. Trying to explore this methodology varying the starting reagents but taking into advantage the one-pot protocol benefits, the study of the three-component reaction of alkyl 2-aminoesters derived from *L*-tryptophan (**2**) and 1,3-dicarbonyl compounds (**3**) under solvent-free conditions towards the synthesis of 2,4,5-trisubstituted-1H-pyrrol-3-ol-type compounds (**1**) is presented herein.

## 2. Results and Discussion

Esterification reactions of *L*-tryptophan were carried out employing Li and Sha methodology [27] with some modifications [28]. Mixtures of *L*-tryptophan (1 eq.), chlorotrimethylsilane (8 eq.) and aliphatic alcohols in excess (R_1_ = Me, Et, *i*-Pr, *n*-Bu) were heated under reflux conditions during 4 h. Alkyl 2-aminoesteres **2a**–**d** were obtained in the hydrochloride form as white crystals in high yields. Then, neutralization with sodium bicarbonate solution and subsequent extractions using isopropyl acetate were performed to obtain compounds **2a**–**d** as the free-base form in a quantitative yield. Several experiments based on **2a**–**d** and 1,3-dicarbonyl compounds **3a**–**d** mixtures were carried out to establish the optimal conditions to afford compounds **1a**–**d**. Thus, several inorganic bases in equimolar ratio regarding compounds **2a**–**d** were initially essayed employing different solvent conditions. Thus, experiments using sodium carbonate in polar solvents under reflux conditions during 24 h afforded polymeric resins, whose chemical characterization attempts were unsuccessful. Similar results were obtained employing potassium hydroxide. However, compounds **1a**–**d** and *L*-tryptophan were detected in low yields (>5%) in all cases. These results suggested that long reaction times, high temperatures and strong basic media favor polymeric and hydrolysis reactions. In this regard, catalytic amount of a strong base could enhance the yield of compound series **1**.

Solvent-free reactions are widely known in literature for their advantages, such as the absence of a reaction medium to collect, purify and recycle high-purity products, the reduction of additional and extensive purification steps, higher yields, low supply of energy, reduction of protection-deprotection steps of functional groups and especially, rapid reaching of substantial reaction completion in few minutes as compared to organic solvent-mediated protocols, among other advantageous features [16]. According to these facts, a solvent-free synthesis under conventional heating conditions was then employed to synthesize the desired products **1a**–**d**, using several essays to explore the performance of the present synthetic protocol. Results are showed in Table 1.

Compound **1a** was obtained from acetylacetone **3a**, which has the highest electrophilic character in comparison with the remaining 1,3-dicarbonyl compounds **3b**–**d**, affording the highest yield (i.e., 81%) when reacted with methyl *L*-tryptophanate **2a** as starting reagent. Aliphatic chain extension of the alkoxy moiety at ester group conduced to a yield reduction, especially using *n*-butyl *L*-tryptophanate **2d**. These results can be explained considering a long-distance inductive effect, which tends to reduce the nucleophilic character of the amine group, affecting the required condition to carry out the first step within the mechanistic pathway. Moreover, these results led us to conclude that the steric implications of the ester fragment are not relevant for the reaction completion, so the low yields were more related to the stability of longer-chain bearing alkoxy moiety, since they degraded more easily and faster (within 5–20 min) and do not lead to the desired cyclization. Similar results were obtained for compounds **1b** and **1d**, i.e., involving low yields, but an identical alkoxy moiety-related trend was also evidenced. Compound **1c** was obtained from malonic acid **3c** used as 1,3-dicarboxylic compound. This reaction series showed yields between 15–45%. To ensure the completion of such reaction series, basic conditions were not employed within the first 5 min and reaction mixtures were submitted to a higher temperature. Therefore, this reaction was performed in two stages: the first one corresponds to the formation of the intermediate amide from malonic acid **3c** and alkyl 2-aminoesters during the first 5 min, and a second stage where a catalytic amount of KOH was added to promote the cyclization reaction towards compound **1c**.

According to these results, a reaction mechanism for the formation **1a**–**d** was proposed (Scheme 2). The first step corresponds to the nucleophilic attack by the amine group in **2** to one of the carbonyl or carboxylic groups in **3** towards the formation of (*Z*)-enamine-type intermediate **4** through a nucleophilic addition-elimination mechanism. Presumably, this first step is the slowest in the mechanistic pathway. Then, hydroxyl ion removes an acidic hydrogen of the intermediate **4** increasing its electronic density to facilitate a 5-*exo-trig* intramolecular cyclization, affording 2,4-dihydro-3*H*-pyrrol-3-ones **5**. Such formation is assisted by the alkoxide release, which abstracts one proton from the formed water molecule in order to restitute the catalytic hydroxyl ion. Finally, intermediate **5** suffers a tautomeric conversion towards **1**, which could be favored by the intramolecular hydrogen bonding formation between hydroxyl group at C3 and the carbonyl group of the substituent at C4 of the pyrrole ring (Figure 1).

Computational calculations were carried out using the DFT-B3LYP method at the 6-31G(d,p) level of theory. Solvation model comprised the isodensity polarizable continuum model (IPCM), which uses a static isodensity surface for the solute cavity [29]. Several authors in the literature have used the Highest Occupied Molecular Orbital (HOMO) Lowest Occupied Molecular Orbital (LUMO) energy gap as a computational descriptor to understand kinetic stability. In general, a large energy gap corresponds to a high energy required for electron excitation [30] and it could be correlated with the nucleophile and electrophile character of reactants in organic reactions. To perform a comparison between the molecular stability of compounds **1** and **5**, computational calculations of HOMO-LUMO were performed. In addition, we presented a description of the hybridization of the HOMO and LUMO orbitals using Natural Bond Orbital (NBO) analysis [31], understanding that a high contribution of *p* orbitals within the molecular orbitals, as a linear combination of atomic orbitals (LCAO), involves a greater planar character and, therefore, a higher tendency to aromaticity of the five-membered heterocyclic ring can be deduced. Frontier molecular orbitals HOMO and LUMO of intermediates **5a**–**d** and compounds **1a**–**d** are presented in Figure 2. HOMO of intermediate **5a** showed an occupancy of 1.88226, and a predominant *p* character (*s*(0.03%), *p* 99.99(99.78%), *d* 6.18(0.19%)), being only localized in two carbon atoms of the benzenoid ring in the indole fragment. Similar results are presented for compound **1a** whose HOMO showed occupancy of 1.89607, but a higher *p* character (*s* (0.23%), *p* 99.99(99.68%), *d* 0.40(0.09%) was observed due to the extended highly-conjugated π system localized on the aromatic indole system. Relatedly, LUMOs for **5a** and **1a** showed appreciable differences: LUMO of intermediate **1a** showed high *p* character (*p* 99.99(97.56%)), being especially located in the C=O π* bond, whereas LUMO of compound **5a** had a lesser *p* character but higher conjugation at pyrrole ring (*p* 9.47(83.13%)). The calculated gap HOMO-LUMO for intermediate **5a** was 0.19893 Hartree; however, compound **1a** afforded 0.14731 Hartree. This difference evidenced a decrease in the LUMO energy once the tautomeric conversion occurred, whose driving force could be based on the aromatic stabilization of the five-membered heterocyclic system.

In order to evaluate the incidence of ester fragment on pyrrole ring stability, HOMO and LUMO analysis was also performed for **1b** and **5b** structures. Intermediate **5b** showed a HOMO with occupancy of 1.93004 and a prevailing *p* character (*s* (10.30%), *p* 8.70 (89.64%), *d* 0.01 (0.06%)). In contrast, HOMO for compound **1b** exhibited an occupancy of 1.84916 and a higher *p* character (*s* (1.13%), *p* 87.69 (98.81%), d 0.06 (0.06%), whose HOMO representation showed the conjugated π system to be also extended towards the pyrrole ring. The LUMO of **5b** showed a higher *p* character (*p* 99.99(97.40%)), and it was located on both C=O ketone and C=O ester π* bonds, whereas LUMO of **1b** was similar to that of **1a** (*p* 10.52 (84.28%)). The calculated HOMO-LUMO gaps of **5b** and **1b** were 0.10324 and 0.16090 Hartree, respectively. In this case, there was no LUMO stabilization for **1b** due to the presence of the ester group. Additionally, the incidence of the hydroxyl group as R^3^ substituent was also examined through a HOMO-LUMO gap analysis. Such an analysis provided gap values for **5c** and **1c** of 0.07948 and 0.12878 Hartree, respectively. The above-mentioned differences between **5b–1b** and **5c–1c** could be explained due a strong inductive and mesomeric effect generated by the presence of polar groups at C2 and C3 of the pyrrole ring. Finally, intermediate **1d** and compound **5d** showed similar DFT-derived results in comparison to the observed outcome for **1a** and **5a** structures.

In order to explain the reaction mechanism, some thermodynamic features were also calculated using DFT-B3LYP methodology. Calculated ΔG° for the conversion of **5**→**1** is presented in Table 2. The conversion of **5** towards **1** is thermodynamically favored in all the studied cases, being a series of exergonic processes (ΔG°_calculated_ < 0). The formation of compound **1b** is presumably a more spontaneous process, while formation of **1a**, **1c** and **1d** are lesser spontaneous. These results supported the discussed hypothesis regarding the formation of intramolecular hydrogen bonding in compounds **1** (Figure 1), whose strength resulted higher when the substituent of the pyrrole ring at C4 is an aliphatic ketone. The inductive effect caused on C=O by the OR group in the alkoxy carbonyl substituent reduces the electronic density of this hydrogen bond acceptor, leading to a decrease of the interaction strength and, consequently, the stability of the aromatic form. In addition, the steric effect of the cyclohexane-type ring fused to the pyrrole system presumably reduces the strength of the hydrogen bonding interaction, affecting the reaction spontaneity.

## 3. Materials and Methods

### 3.1. General Information

All chemicals were purchased from Sigma-Aldrich (Saint Louis, MO, USA) and Merck KGaA (Darmstadt, Germany) and used without further purification. Dry solvents were purchased in sufficient purity. Thin layer chromatography (TLC) was done on TLC silica gel 60 F254 (Merck KGaA), and compounds were detected at 254 nm. Column chromatography was conducted manually on silica gel 60 (0.040–0.063 mm) from Merck KGaA. Nuclear magnetic resonance (NMR) spectra were recorded on a Bruker Avance AV-400 MHz spectrometer (Billerica, MA, USA). All shifts are given below in δ (ppm) using the signal of tetramethylsilane (TMS) as a reference. All coupling constants (*J*) are given in Hz. Splitting patterns are typically described as follows: *s*: singlet, *d*: doublet, *t*: triplet, *q*: quartet and *m*: multiplet. Liquid chromatography-mass spectrometry (LC/MS) experiments were performed on a LCMS 2020 spectrometer (Shimadzu, Columbia, MD, USA), comprising a Prominence^®^ high-performance liquid chromatography (HPLC) system coupled to a quadrupole single analyzer with an electrospray ionization (ESI). A Synergi^®^ column (150 × 4.6 mm, 4.0 µm) was used for analysis at 0.6 mL/min using mixtures of acetonitrile (A) and 1% formic acid (B) in gradient elution. ESI was operated simultaneously in positive and negative ion modes (scan 100–2000 *m/z*), desolvation line temperature at 250 °C, nitrogen as nebulizer gas at 1.5 L/min, drying gas at 8 L/min, and detector voltage at 1.4 kV. Accurate mass data were recorded by high-resolution MS (HRMS) on a micrOTOF-Q II mass spectrometer (Bruker, Billerica, MA, USA). ESI was also operated in positive and negative ion modes (scan 100–2000 *m/z*), desolvation line temperature at 250 °C, nitrogen as nebulizer gas at 1.5 L/min, drying gas at 8 L/min, quadrupole energy at 7.0 eV, and collision energy 14 eV.

### 3.2. General Experimental Procedure for the Synthesis of 2,4,5-trisubs-tituted-1H-pyrrol-3-ol Type Compounds (***1a***,***b***,***d***)

Alkyl 2-aminoesters **2** (1 mmol), 1,3-dicarbonyl compounds **3** (1 mmol), and potassium hydroxide (0.1 mmol) mixtures were heated in open vessels employing an oil bath with shaking following the reaction conditions in Table 1. TLC monitored the progress of each reaction. Upon completing the reaction, 5 mL of saturated ammonium chloride solution was added to each resulting reaction mixture. Discontinuous liquid-liquid extractions were performed using chloroform (5 × 10 mL). Reduced pressure was used for concentrating the reaction mixtures. Conventional column chromatography using silica gel as sorbent and a hexane: ethyl acetate mixtures as mobile phase was employed for purifying the afforded products **1a**–**d**. The structure of each compound was confirmed by analytical as well as conventional spectral studies (Appendix A).

#### Representative Compounds

1-(5-((1*H*-indol-3-yl)methyl)-4-hydroxy-2-methyl-1*H*-pyrrol-3-yl)ethan-1-one (**1a**): Yellow oil; yield: 15–81%. ^1^H NMR (400 MHz, CDCl_3_): δ = 9.33 (*s*, 1H), 7.58 (*m*, 1H), 7.32 (*m*, 1H), 7.11 (*m*, 1H), 7.10 (*m*, 2H), 3.32 (*s*, 2H), 2.03 (*s*, 3H), 1,68 (*s*, 3H) ppm. ^13^C NMR (100 MHz, CDCl_3_): δ = 195.6, 162.15, 136.33, 127.02, 124.21, 121.68, 121.39, 119.11, 118.14, 117.99, 111.56, 111.08, 108.91, 96.48, 69.36, 29.45, 26.55 ppm. ESI-MS in the positive mode *m/z*: [M + H]^+^, 269.16; [M + CH_3_CN + H]^+^, 310.19. ESI-HRMS *m/z*: [M + H]^+^, 269.1677 (calcd. 269.1690).

Ethyl 5-((1*H*-indol-3-yl)methyl)-4-hydroxy-2-methyl-1*H*-pyrrole-3-carboxylate (**1b**): Yellow oil; yield: 19–54%. ^1^H NMR (400 MHz, CDCl_3_): δ = 8.45 (*s*, 1H), 7.57 (*m*, 1H), 7.31 (*m*, 1H), 7.16 (*m*, 2H), 7.09 (*s*, 1H), 4.10 (*m*, 2H), 4.08 (*s*, 3H), 3.31 (*s*, 2H), 1.68 (*s*, 3H), 0.88 (*t*, *J* = 6.0 Hz, 3H) ppm. ^13^C NMR (100 MHz, CDCl_3_): δ = 172.26, 160.38, 136.23, 127.12, 123.72, 122.23, 122.03, 119.66, 119.46, 118.32, 111.45, 109.76, 109.58, 109.50, 65.40, 19.31, 14.64, 13.71 ppm. ESI-MS *m/z* in the positive mode: [M + H]^+^, 299.12; [M + CH_3_CN + H]^+^, 339.23. ESI-HRMS *m/z*: [M − H]^−^, 297.1266 (calcd. 279.1239).

2-((1*H*-indol-3-yl)methyl)-3-hydroxy-1,5,6,7-tetrahydro-4*H*-indol-4-one (**1d**): Yellow oil; yield: 17–59%. ^1^H NMR (400 MHz, CDCl_3_): δ = 9.34 (*s*, 1H), 7.46 (*m*, 1H), 7.33 (*m*, 1H), 7.11 (*m*, 1H), 7.05 (*m*, 1H), 6.93 (*s*, 1H), 3.33 (*s*, 2H), 2.29 (*m*, 2H), 2.20 (*m*, 2H), 1.88 (*m*, 2H) ppm. ^13^C NMR (100 MHz, CDCl_3_): δ = 171.01, 163.68, 136.23, 127.72, 123.26, 121.91, 121.91, 119.31, 119.31, 118.23, 111.56, 108.85, 97.44, 60.45, 36.33, 21.69, 14.17 ppm. ESI-MS in the positive mode *m/z*: [M + H]^+^, 281.16. ESI-HRMS *m/z*: [M + H]^+^, 279.1134 (calcd. 279.1135).

### 3.3. General Experimental Procedure for the Synthesis of 5-((1H-indol-3-yl)methyl)-2,4-dihydroxy-1H-pyrrole-3-carboxylic Acid (***1c***)

Alkyl 2-aminoesters **2** (1 mmol) and malonic acid **3c** (1 mmol) were heated in open vessels employing an oil bath at 170 °C for 5 min. Potassium hydroxide (0.1 mmol) was added to the reaction mixture, and the heating was kept at 170 °C for 25 min. Then, 5 mL of saturated ammonium chloride solution was added to each resulting reaction mixture. Discontinuous liquid-liquid extractions were performed using chloroform (5 × 10 mL). Reduced pressure was used for concentrating the reaction mixtures. Conventional column chromatography using silica gel as sorbent and hexane: ethyl acetate mixtures as mobile phase was employed for purifying the afforded product **1c**: Yellow oil; yield: 15–45%. ^1^H NMR (400 MHz, CDCl_3_): δ = 8.24 (*s*, 1H), 7.52 (*m*, 1H), 7.35 (*m*, 1H), 7.13 (*m*, 2H), 6.97 (*s*, 1H), 3.70 (*s*, 2H) ppm. ^13^C NMR (100 MHz, CDCl_3_): δ = 172.59, 169.94, 169.94, 136.24, 136.24, 127.89, 119.86, 119.86, 118.66, 111.44, 111.44, 110.21, 110.21, 23.36 ppm. ESI-MS in the positive mode *m/z*: [M + H]^+^, 273.15; [M + CH_3_CN + H]^+^, 313.11.

## 4. Conclusions

In conclusion, we developed a conveniently efficient and environmentally benign alternative protocol for easy access to a series of diverse 2,4,5-trisubstituted-1*H*-pyrrol-3-ol type compounds from the one-pot three-component reaction between alkyl 2-aminoesters derived from *L*-tryptophan, potassium hydroxide and 1,3-dicarbonyl compounds in solvent-free conditions. The compounds **1a**–**d** were obtained in moderate to good yields in short reaction times, operational simplicity, and an easy workup. DFT-B3LYP calculations demonstrated the preference of the performed protocol towards 1*H*-pyrrol-3-ol aromatized rings, which are formed through the tautomeric equilibrium from 3*H*-pyrrol-3-ones.

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
