# Peer review of "Solvent Free Three-Component Synthesis of 2,4,5-trisubstituted-1H-pyrrol-3-ol-type Compounds from L-tryptophan: DFT-B3LYP Calculations for the Reaction Mechanism and 3H-pyrrol-3-one↔1H-pyrrol-3-ol Tautomeric Equilibrium"

_molecules, 2020, doi:10.3390/molecules25194402_

Round 1
Reviewer 1 Report
The manuscript entitled “Solvent free three-component synthesis of 2,4,5-trisubstituted-1H-pyrrol-3-ol type compounds from L-tryptophan: DFT-B3LYP calculations for the reaction mechanism and 3H-pyrrol-3-one: 1H-pyrrol-3-ol tautomeric equilibrium” by Quiroga et. al. explores the synthesis of novel indole-containing pyrrol-3-ol compounds. I think these structures are sufficiently interesting, and the method straightforward enough to warrant publication in Molecules, though I do have a few concerns to address first:
- The manuscript needs significant proofreading/copyediting. There are minor typos, article omissions, etc. that make it difficult to read.
- Table 1 isn’t very useful as presented. I assume the differences in some cases are due to the R1 group, but this needs to be more clearly demonstrated. Furthermore, calling the product the entry is not really accurate. Entries should be independently numbered.
- In Scheme 2, the product 1 doesn’t have any remaining chirality, as it is planar.
- The reaction of 4 to give 5 should involve loss of the alcohol/alkoxide, so this could be clarified (OH is catalytic, with the proton actually ending up on the more basic alkoxide byproduct).
- The computational section doesn’t really seem to fit into the manuscript. I don’t understand what it is trying to address – that the aromatic tautomer is favored seems obvious, and I believe would be readily predicted without the need for DFT (I think ChemSketch by ACD labs for example can readily provide this information), so doing the DFT seems odd.
- I also feel like more information is needed to explain the DFT, such as what solvation model is used.
- Ref 29 is not listed in the references section, but is included in the text (I think this is a typo and was meant to be Ref 28 in the text, but the authors should check this).
- On line 195 “Conventional CC” needs to be defined. I assume the authors mean column chromatography over silica gel, but this needs to be clearly stated. Furthermore, solvent conditions should be noted.
- Compounds 1a – 1d are all new. It is therefore appropriate to include the 1H and 13C spectra to demonstrate purity.
- High Resolution Mass Spectrometry data is usually used to convey both identity and to establish purity (though I disagree that it does the latter). However, the data presented seem to have very high error, and generally HRMS data should have 4 decimal places, which is another issue. The authors should both provide the calculated and observed masses, and these should really be within 5 parts per million error, as is standard in reporting new organic molecules. Alternatively, using the standards of Org. Chem. the MS data can be substituted for elemental analysis.
- Related also to the MS data, can the authors explain how the [M + CH3CN]+ adduct forms? This is not a method for ionization that I am familiar with. In the cas of 1a it appears to have been incorrectly determined, as 310 would actually be [M + H + CH3CN]+.
Author Response
Reviewer 1
Dear Reviewer.
Thanks for all the comments. We send a revised and corrected version of the manuscript, wherein were performed modifications according to your suggestions and comments, which are answered in the following paragraphs:
- The manuscript needs significant proofreading/copyediting. There are minor typos, article omissions, etc. that make it difficult to read.
Answer: Thank you for your remark. Proofreading and copyediting were then performed. Two complete revisions were carried out by means of a grammatical software and the revision of an English writing expert. You can check all the changes in the revised version.
- Table 1 isn’t very useful as presented. I assume the differences in some cases are due to the R1 group, but this needs to be more clearly demonstrated. Furthermore, calling the product the entry is not really accurate. Entries should be independently numbered.
Answer: Compounds are labeled by bold numbers according to the compound type in the manuscript. If a specific compound of any type is referred, a letter indicates such a particular compound. For instance, compound 1a is a pyrrole derivative obtained from acetylacetone 3a and different alkyl 2-aminoesters. Therefore, concerning table 1, we accept your comments. There were many inaccuracies related to the substituent but especially with the reaction conditions. So, Table 1 was corrected.
- In Scheme 2, the product 1 doesn’t have any remaining chirality, as it is planar.
Answer: We accept your comment. Scheme 2 was corrected, removing the chirality of the compound 1.
- The reaction of 4 to give 5 should involve loss of the alcohol/alkoxide, so this could be clarified (OH is catalytic, with the proton actually ending up on the more basic alkoxide byproduct).
Answer: We accept your comment. The paragraph where is explained the reaction mechanism, line 137, now appears like this: “Then, hydroxyl ion removes an acidic hydrogen of the intermediate 4 increasing its electronic density to facilitate a 5-exo-trig intramolecular cyclization, affording 2,4-dihydro-3H-pyrrol-3-ones 5. Such formation is assisted by the alkoxide release, which abstracts one proton from the formed water molecule in order to restitute the catalytic hydroxyl ion.”
- The computational section doesn’t really seem to fit into the manuscript. I don’t understand what it is trying to address – that the aromatic tautomer is favored seems obvious, and I believe would be readily predicted without the need for DFT (I think ChemSketch by ACD labs for example can readily provide this information), so doing the DFT seems odd.
Answer: We thank your kind remark but we do not totally agree with your comment. Many reactions in literature are affected by several electronic and steric effects, which have consequences in their molecular structure and the reaction pathway. In this regard, our initial purpose was to obtain compound series 1 through the formation of an imine intermediate. We collected scientific evidence that led us to conclude imine form is not favored, driving the reaction to enamine form in the specific Z isomer, which is not expected for this kind of compound. DFT calculations are exciting tools to perform studies about the reaction mechanism and the differences between intermediate and product with tautomeric forms. Due to these arguments and our experience in this kind of reaction, we decided to employ these computational tools, which are abundant in literature but using only two of them: the NBO analysis, which gives us information about the molecular orbitals (HOMO and LUMO in the manuscript) and ΔG° of each reaction. Accordingly, we included the following paragraph in line 150: “Computational calculations were carried out using the DFT method at the B3LYP/6-31G(d,p) level of theory. Solvation model comprised the isodensity polarizable continuum model (IPCM), which uses a static isodensity surface for the solute cavity [29]. Several authors in the literature have used the HOMO-LUMO energy gap as a computational descriptor to understand kinetic stability. In general, a large energy gap corresponds to a high energy required for electron excitation [30] and it could be correlated with the nucleophile and electrophile character of reactants in organic reactions.” In our criterion, these results allowed us to understand in a better form our mechanistic purpose and the stability of compounds 1.
- I also feel that more information is needed to explain the DFT, such as what solvation model is used.
Answer: We accept your comment. Starting from line 150, we included this information.
- Ref 29 is not listed in the references section, but is included in the text (I think this is a typo and was meant to be Ref 28 in the text, but the authors should check this).
Answer: We accept your comment. The correction about references 28 and 29 were performed in the corrected version of the manuscript.
- On line 195 “Conventional CC” needs to be defined. I assume the authors mean column chromatography over silica gel, but this needs to be clearly stated. Furthermore, solvent conditions should be noted.
Answer: We accept your comment. Complete and precise information about column chromatography was included in line 236.
- -Compounds 1a – 1d are all new. It is therefore appropriate to include the 1H and 13C spectra to demonstrate purity.
Answer: We accept your comment. We included supplementary material which contains 1H and 13C NMR spectra of compounds 1a-d.
- High Resolution Mass Spectrometry data is usually used to convey both identity and to establish purity (though I disagree that it does the latter). However, the data presented seem to have very high error, and generally HRMS data should have 4 decimal places, which is another issue. The authors should both provide the calculated and observed masses, and these should really be within 5 parts per million error, as is standard in reporting new organic molecules. Alternatively, using the standards of Org. Chem. the MS data can be substituted for elemental analysis.
Answer: We accept your comment partially. We sent samples to another lad outside country of compounds 1a-d to perform HRMS before the quarantine for the COVID-19 pandemic. The original submission included mass spectrometry data obtained from HPLC-MS analysis, where the ionization technique is electrospray ionization. The results of HRMS were received the last week, so we included them in the final version. Unfortunately, the sample 1c mass spectrometry data were not successful due to the compound low stability.
- Related also to the MS data, can the authors explain how the [M + CH3CN]+ adduct forms? This is not a method for ionization that I am familiar with. In the cas of 1a it appears to have been incorrectly determined, as 310 would actually be [M + H + CH3CN]+.
Answer: We accept your comment. The signal as m/z 310 corresponds to the [M+CH3CN+H+]+ adduct. As we mentioned in the last reply, we performed HPLC-MS analysis, whose mass analyzer included electrospray ionization. This technique produces ions using an electrospray in which a high voltage is applied to a liquid to create an aerosol. It is especially useful for producing ions from macromolecules because it overcomes the fragmentation susceptibility of these molecules when ionized. Due to this phenomenon, the mass spectrometry data led us to measure m/z relationships, which can be attributed to adducts formation between the solvent and the analyte. In our specific case, the mobile phases used for HPLC-MS were acetonitrile: water mixtures. So, the results for compounds 1a-c showed the formation of these kinds of adducts.
Reviewer 2 Report
The authors described the synthesis of 2,4,5-trisubstituted-1H-pyrrolo-3-ols under solvent-free conditions. The authors also investigated on the reaction mechanism. The reaction will be able to be achieved conventionally. And the products were novel. Thus, I can accept this manuscript to publish in Molecules. But I recommend revising this manuscript on the survey of the reaction conditions and the discussion of the reaction mechanism based on the comments as follows.
1) How to determine the reaction temperature? For example, the formation of 1a was conducted at 100 ℃, but the reaction for 1b was raised the temperature to 120 ℃. Especially, the treatment at high temperature (170 ℃) was used in the case of formation of 1c. Please explain those differences. In addition, the authors commented the reaction to give 1c will be spontaneous process in line 166. But the high temperature was required. How to explain the disagreement of the result and the postulation.
2) In line 106, the authors wrote the formation of 1a in 85% yield. However, I could not find the data in Table 1. Please check it.
3) In Table 1, I could not understand the difference of the conditions in some entries (for instance, first line (81% yield) and second line (24% yield)) although the difference would be arisen from the structure of 2. Please clarify those differences like as the notation for 2 or R1. In addition, the entry should be numbering to distinguish each experiment. Thus, it is curious to exist the same entry number.
4) The authors commented the necessary to avoid basic conditions in lines 114-115. However, no specific treatment was found in the experimental procedure. How should the readers care for the problem when the readers conduct this procedure? Please show the detailed technical information to accommodate the reaction smoothly.
5) The authors discussed the stability of 1 and 5 based on HOMO/LUMO and ΔG. But the energetical difference is important to conclude the stability. Thus, I think that the discussion about ΔG is enough to explain the stable of 1 in this manuscript.
Focused on ΔG, TS was shown in Figure 2. But the tautomerism from 5 to 1 includes two isomerizations (ketone to enol, and imine to enamine). Do the authors want to explain that those isomerizations occurs at once? If so, I cannot accept that allegation based on the fact shown in this manuscript. I recommend showing the value of ΔG of 5 and 1 in each compound with the difference value of ΔG. In addition, the authors commented the influence of hydrogen bond to distinguish the difference of ΔG (lines 164-168). Is there hydrogen bond in 1c in the structure depicted in Figure 1? Please discuss the hydrogen bond based on the optimized structure given ΔG.
6) About the comment of molecular orbitals, I could not accept the authors’ explanation about the decease of LUMO energy based on the small difference between HOMO and LUMO energies. This phenomenon should be explained based on the each LUMO energy which must be obtainable.
I could not understand “the p character”. I think the hybridization character should be represented in each atom not in molecule. How to find those values in the calculation data?
Additional trivial matter: I think “1b” in line 136 seems to be “1a”, and “5a” in line 138 seems to be “1a”. Please check them.
Additional requirement.
Reference numbers did not match (for example, line 79 [27] should be [26], line 131 [29] may be [27] and/or [28]
Author Response
Reviewer 2
Dear Reviewer.
Thanks for all the comments. We send a revised and corrected version of the manuscript, wherein were performed modifications according to the suggestions and comments which are answered in the following paragraphs:
1) How to determine the reaction temperature? For example, the formation of 1a was conducted at 100 ℃, but the reaction for 1b was raised the temperature to 120 ℃. Especially, the treatment at high temperature (170 ℃) was used in the case of formation of 1c. Please explain those differences. In addition, the authors commented the reaction to give 1c will be spontaneous process in line 166. But the high temperature was required. How to explain the disagreement of the result and the postulation.
Answer: The reaction temperature was established in preliminary experiments using HPLC-MS for each reaction mixture using heating at 100, 120, and 170 °C during 5 min and evidencing product formation and reactant consumption. Each compound was formed in a different temperature condition. Then, we performed all the essays varying the 2-amino ester substrates. Unfortunately, Table 1 had many inaccuracies in the submitted version of the manuscript. We present a revised and corrected version of the manuscript, including several changes in Table 1 to allow an easy and clear comprehension. According to the term “spontaneous”, it refers to the 5→1 conversion but not related to the formation of compounds 1, which need heating to be formed.
2) In line 106, the authors wrote the formation of 1a in 85% yield. However, I could not find the data in Table 1. Please check it.
Answer: We accept your comment. A corrected version of Table 1 was included in the manuscript.
3) In Table 1, I could not understand the difference of the conditions in some entries (for instance, first line (81% yield) and second line (24% yield)) although the difference would be arisen from the structure of 2. Please clarify those differences like as the notation for 2 or R1. In addition, the entry should be numbering to distinguish each experiment. Thus, it is curious to exist the same entry number.
Answer: We accept your comment. A corrected version of Table 1 was included in the manuscript.
4) The authors commented the necessary to avoid basic conditions in lines 114-115. However, no specific treatment was found in the experimental procedure. How should the readers care for the problem when the readers conduct this procedure? Please show the detailed technical information to accommodate the reaction smoothly.
Answer: Performed essays to afford compound 1c were unsuccessfully when basic conditions were used from the starting materials. We noted that basic conditions promote decarboxylation reaction on both malonic acid and a possible amide intermediate. So, we performed the reaction in two stages: the first one corresponds to the formation of the intermediate amide from malonic acid and 2-amino esters during the first 5 minutes and a second stages where a catalytic amount of KOH was added to promote the cyclization reaction towards compound 1c with low-to-moderate yields.
5) The authors discussed the stability of 1 and 5 based on HOMO/LUMO and ΔG. But the energetical difference is essential to conclude the stability. Thus, I think that the discussion about ΔG is enough to explain the stable of 1 in this manuscript.
Answer: DFT calculations are exciting tools to perform studies about the reaction mechanism and the differences between intermediate and product tautomeric forms. Due to these arguments and our experience in this kind of reaction, we decided to employ these computational tools, which are abundant in literature but using only two of them: the NBO analysis, which gives us information about the molecular orbitals (HOMO and LUMO in the manuscript) and DG° of each reaction. We include the following paragraph in line 143: “Computational calculations were carried out using the DFT method at the B3LYP/6-31G(d,p) level of theory. Solvation model comprised the isodensity polarizable continuum model (IPCM), which uses a static isodensity surface for the solute cavity [29]. Several authors in the literature have used the HOMO-LUMO energy gap as a computational descriptor to understand kinetic stability. In general, a large energy gap corresponds to a high energy required for electron excitation [30] and it could be correlated with the nucleophile and electrophile character of reactants in organic reactions.”
6) Focused on ΔG, TS was shown in Figure 2. But the tautomerism from 5 to 1 includes two isomerizations (ketone to enol, and imine to enamine). Do the authors want to explain that those isomerizations occurs at once? If so, I cannot accept that allegation based on the fact shown in this manuscript. I recommend showing the value of ΔG of 5 and 1 in each compound with the difference value of ΔG. In addition, the authors commented the influence of hydrogen bond to distinguish the difference of ΔG (lines 164-168). Is there hydrogen bond in 1c in the structure depicted in Figure 1? Please discuss the hydrogen bond based on the optimized structure given ΔG.
Answer: Your comment is right. Thanks. Figure 2 pretended to give to readers a graphical idea about the 5→1 conversion. These transformations only can succeed towards two transition states corresponding to the isomerization reactions mentioned by you. To include correct information, Figure 2 was removed from the revised manuscript. We included a new table labeled as “Table 2”, which contains calculated DG° in kcal/mol for the conversions 5→1. Concerning the hydrogen bond proposed, a new Figure labeled as “Figure 1” was included in the manuscript, which represents the DFT-optimized molecular structure of the compound 1a and the hydrogen bonding formed between the hydrogen atom of the OH group at C3 and the carbonyl group as substituent at C4. Moreover, it is known that the hydrogen bond length usually is between 1.5-2.5 Å. The H…O length in compounds 1a-d, calculated from the optimized molecular structures using DFT B3LYP at the level 6-31G(d,p), was found in this range, which can be used as computational evidence of the proposed hydrogen bond.
7) About the comment of molecular orbitals, I could not accept the authors’ explanation about the decease of LUMO energy based on the small difference between HOMO and LUMO energies. This phenomenon should be explained based on the each LUMO energy which must be obtainable.
Answer: Calculations performed for HOMO and LUMO were explained detail in the manuscript's revised version. These calculations are used for several authors in the literature as a computational descriptor to understand the kinetic stability in organic compounds. In general, a large energy gap corresponds to a high energy required for electron excitation and it could be correlated with the nucleophile and electrophile organic reactions. So, we used these results to explain the stability differences between compounds 1 and 5.
8) I could not understand “the p character”. I think the hybridization character should be represented in each atom not in molecule. How to find those values in the calculation data?
Answer: The p character of the HOMO and LUMO for compounds 1 and 5 were used to explain that a high contribution of p orbitals within the molecular orbitals as a linear combination of atomic orbitals (LCAO), involves a more significant planar character, therefore a higher tendency to aromaticity of the five-membered heterocyclic ring is deduced. It was obtained employing the NBO analysis, which is based on a method for optimally transforming a given wave function into localized form, corresponding to the one-center ("lone pairs") and two-center ("bonds") elements of the chemical structure. In NBO analysis, the input atomic orbital basis set is transformed via natural atomic orbitals (NAOs) and natural hybrid orbitals (NHOs) into natural bond orbitals (NBOs). For each of the NAO functions, calculation results are presented as a table, where the atom is listed to which the NAO is attached, including the angular momentum type 'lang' (s, px, etc.), the orbital type (whether core, valence, or Rydberg, and a conventional hydrogen-type label), the orbital occupancy (number of electrons, or 'natural population' of the orbital), and the orbital energy (in Hartree).
9) Additional trivial matter: I think “1b” in line 136 seems to be “1a”, and “5a” in line 138 seems to be “1a”. Please check them.
Answer: We accept your comment. The information containing in both lines was corrected.
Additional requirement.
10) Reference numbers did not match (for example, line 79 [27] should be [26], line 131 [29] may be [27] and/or [28]
Answer: We accept your comment. The correction about references 28 and 29 were performed in the corrected version of the manuscript.
Round 2
Reviewer 1 Report
N/A